# Herlyn–Werner–Wunderlich Syndrome Complicated with Vesicovaginal Fistula: A Rare Case Report

**DOI:** 10.3390/medicina60071081

**Published:** 2024-07-01

**Authors:** Ruei-Lin Wang, Yu-Kuen Wang, Chen-Hsien Lin, Jenq-Shyong Chan, Hang-Seng Liu, Po-Jen Hsiao

**Affiliations:** 1Division of Nephrology, Department of Internal Medicine, Taoyuan Armed Forces General Hospital, Taoyuan 325, Taiwan; aassukw@livemail.tw (R.-L.W.); jschan0908@yahoo.com.tw (J.-S.C.); 2Division of Nephrology, Department of Internal Medicine, Tri-Service General Hospital, National Defense Medical Center, Taipei 114, Taiwan; 3Division of Obstetrics and Gynecology, Taoyuan Armed Forces General Hospital, Taoyuan 325, Taiwan; yukuen0922@aftygh.gov.tw (Y.-K.W.); bill13@aftygh.gov.tw (C.-H.L.); 0912992335@icloud.com (H.-S.L.); 4Division of Nephrology, Department of Internal Medicine, Tri-Service General Hospital Songshan Branch, National Defense Medical Center, Taipei 114, Taiwan; 5Department of Life Sciences, National Central University, Taoyuan 320, Taiwan

**Keywords:** Herlyn–Werner–Wunderlich syndrome, uterus didelphys, renal agenesis, vesicovaginal fistula

## Abstract

Herlyn–Werner–Wunderlich (HWW) syndrome is characterized by obstructed hemivagina and ipsilateral renal anomaly, a rare congenital anomaly of the genitourinary tract, resulting from malformations of the renal tract associated with Müllerian duct anomalies. The initial symptoms of HWW frequently present after menarche and may be nonspecific, leading to a delayed diagnosis. We presented a 19-year-old female with 3-year hematuria and abdominal pain. The final diagnosis of HWW syndrome with a rare vesicovaginal fistula was made. The treatment of HWW syndrome typically involves surgical intervention. The primary treatment is resection or removal of the obstructed vaginal septum. The patient underwent excision of vaginal septum and vaginal reconstruction via hysteroscopy, as well as repair of the vesicovaginal fistula. The patient improved well after surgery and fully recovered without sequelae after 3 months. In addition, unilateral renal agenesis is one of congenital abnormalities of the kidney and urinary tract, which are the most frequent cause of chronic kidney disease (CKD) in children. This report describes a patient of HWW syndrome with rarely combined vesicovaginal fistula, and highlights the importance of early recognition and management to prevent associated complications.

## 1. Introduction

Herlyn–Werner–Wunderlich (HWW) syndrome is a rare congenital disorder affecting the female reproductive system. Named after the physicians who first described it in 1971, this condition involves a combination of Müllerian duct anomalies, typically presenting as a triad of uterus didelphys, obstructed hemivagina, and ipsilateral renal agenesis [1]. The development of the female reproductive system begins with the fusion of two Müllerian ducts, which later differentiate into various structures, including the uterus, fallopian tubes, and upper vagina. In HWW syndrome, a defect occurs during this process, resulting in the incomplete fusion of the Müllerian ducts. This leads to the formation of a double uterus (uterus didelphys), with each uterus connected to its respective fallopian tube [2]. Additionally, one side of the vagina may be obstructed, causing the accumulation of menstrual blood in the obstructed hemivagina. Meanwhile, the ipsilateral kidney (on the same side as the obstructed hemivagina) is often absent or underdeveloped due to abnormal embryonic development [2].

The symptoms of HWW syndrome usually become apparent after puberty, when the individual begins to menstruate. The diagnosis of HWW syndrome typically depends on a combination of imaging techniques such as ultrasound, magnetic resonance imaging (MRI), and sometimes computed tomography (CT) scans [3]. These imaging modalities help visualize the abnormalities in the reproductive and urinary systems, confirming the presence of uterine duplication, obstructed hemivagina, and renal anomalies.

Management of HWW syndrome usually involves surgical intervention to relieve the obstruction in the hemivagina and prevent complications such as recurrent infections and endometriosis. It typically requires excision of the obstructed vaginal septum to allow proper drainage of menstrual blood [4].

## 2. Case Presentation

A 19-year-old female with no past medical history, no history of pregnancy, and no long-term medication use presented with recurrent abdominal pain and bloody urine persisting for 3 years. On physical examination, she was afebrile but with tenderness in the lower abdomen. The result of urinalysis revealed pyuria (white blood cell > 100/high power field, HPF), bacteriuria (3+), and hematuria (red blood cell > 100/HPF). She was treated with oral antibiotics, and the subsequent urine culture result was *Escherichia coli*. A trans-abdominal sonogram showed uterus didelphys (Figure 1A) with a vaginal hematoma of 9.24 × 4.6 square centimeters (Figure 1B). Abdominal computed tomography revealed right renal agenesis and left hydronephrosis (Figure 2A). Pelvic magnetic resonance imaging (MRI) demonstrated uterus didelphys, double vagina, right-sided vaginal hematoma, and a vesicovaginal fistula (Figure 2B,C). The final diagnosis of Herlyn–Werner–Wunderlich (HWW) syndrome with a rare and complicated vesicovaginal fistula was made. Ultimately, the patient underwent an excision of the vaginal septum and vaginal reconstruction via hysteroscopy (Figure 3), as well as repair of the vesicovaginal fistula. During the procedure, an engorged right vaginal wall was noted, along with pus and hematoma in the right vaginal diverticulum. A speculum was inserted into the vagina for the excision of the diverticulum. Resectoscopic resection of the diverticulum and drainage were performed. Marsupialization of the right vaginal wall was performed, followed by a Foley catheter insertion into the diverticulum for drainage. Blood loss was about 30 mL. The patient had a good postoperative recovery and was discharged successfully on the second day after surgery. After a 3-month follow-up, the patient had a complete recovery without any sequelae.

## 3. Discussion

HWW syndrome is characterized by an ipsilateral renal anomaly, an obstructed hemivagina, and a rare congenital anomaly of the genitourinary tract. The etiology of HWW syndrome remains unclear, although it is believed to stem from abnormalities during embryonic development [5]. During fetal development, the Müllerian ducts differentiate to form the female reproductive organs, including the uterus, fallopian tubes, and vagina. In cases of HWW syndrome, the failure of Müllerian duct fusion leads to the formation of a uterus didelphys, accompanied by a hemivagina on one side. Additionally, ipsilateral renal agenesis is thought to be a consequence of the same developmental anomaly. Few cases have reported acute urinary retention and hydronephrosis as late complications [6]. Diagnosis usually involves a combination of clinical evaluation and imaging studies. Transabdominal or transvaginal ultrasound is often used as the first-line imaging modality to assess pelvic anatomy and detect the presence of a double uterus, obstructed hemivagina, and ipsilateral renal agenesis, while pelvic MRI can provide more detailed information [7]. One case report described a rare case of HWW syndrome complicated by a urethrovaginal fistula. The patient ultimately underwent surgical treatment to manage the problem of the fistula [8]. Surgical intervention is typically required to manage HWW syndrome. The primary management involves resection or removal of the obstructed vaginal septum to relieve the obstruction. This procedure helps normalize menstrual flow and reduces menstrual pain and infection risk.

Extending from the above issue, unilateral renal agenesis (URA) is one of the congenital abnormalities of the kidney and urinary tract (CAKUT), which is the most frequent cause of chronic kidney disease in children. While single-gene defects are likely responsible for many types of CAKUT, mutations in only a few genes have been identified [9]. The reported prevalences of these anomalies are highly variable. According to an analysis based on 43 included studies, the general incidence of URA was about 1 in 2000, of which vesicoureteral reflux was most frequently identified, and it might be associated with non-obstructive hydronephrosis. Assessment of URA prevalence could be important, as it may help clinicians to ascertain a general and renal prognosis for patients with URA. URA may be a more-or-less harmless congenital malformation; however, a previous study demonstrated that 40–50% of adults with URA required dialysis by the age of 30 years. Though reported in a selected series of URA patients and thus likely overestimating the true risk of kidney disease progression, this impaired outcome may be explained by the hyperfiltration hypothesis [10]. This case highlighted the diagnostic and therapeutic approaches among these patients and emphasized paying more attention to the associated complications. URA is not a harmless malformation by definition, and clinical follow-ups on kidney function are important. Furthermore, the collaboration between nephrologists, urologists, and gynecologists for the management of patients with CAKUT is invaluable. They could work together to provide comprehensive care for patients with CAKUT, including bladder dysfunction, urinary tract obstruction, urinary tract infection, and even vesicovaginal fistula. This partnership enables the prevention and effective management of these complications, leading to improved outcomes for individuals with CAKUT [11].

## 4. Conclusions

Herlyn–Werner–Wunderlich syndrome is a rare but significant condition characterized by a triad of uterine, vaginal, and renal anomalies. In this report, we presented a case of HWW syndrome with unusual vesicovaginal fistula. Last but not least, early recognition and appropriate management are crucial in preventing long-term complications, and clinical follow-up on kidney function is important.

## Figures and Tables

**Figure 1 medicina-60-01081-f001:**
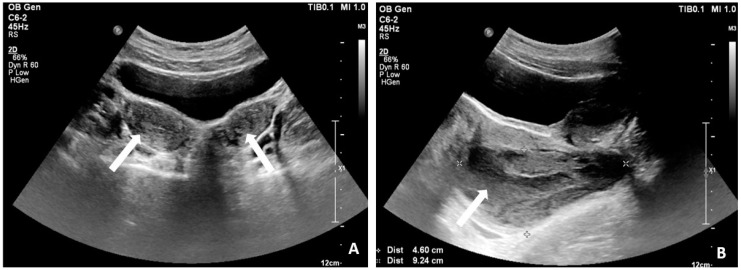
(**A**) uterus didelphys (arrows); (**B**) a vaginal hematoma (arrow) in size of 9.24 × 4.6 square centimeters below the uterus.

**Figure 2 medicina-60-01081-f002:**
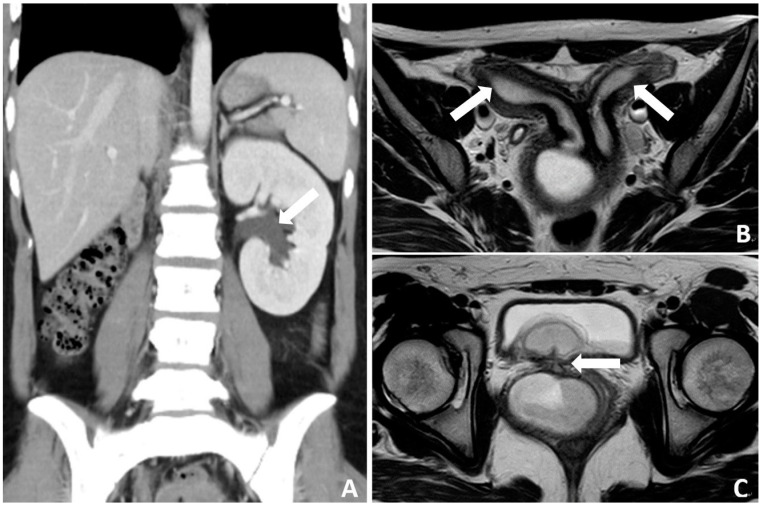
(**A**) abdominal CT revealed left hydroureteronephrosis (arrow) and agenesis right kidney; (**B**) pelvic MRI demonstrated uterus didelphys (arrows) connecting separated vagina; (**C**) vesicovaginal fistula (arrow).

**Figure 3 medicina-60-01081-f003:**
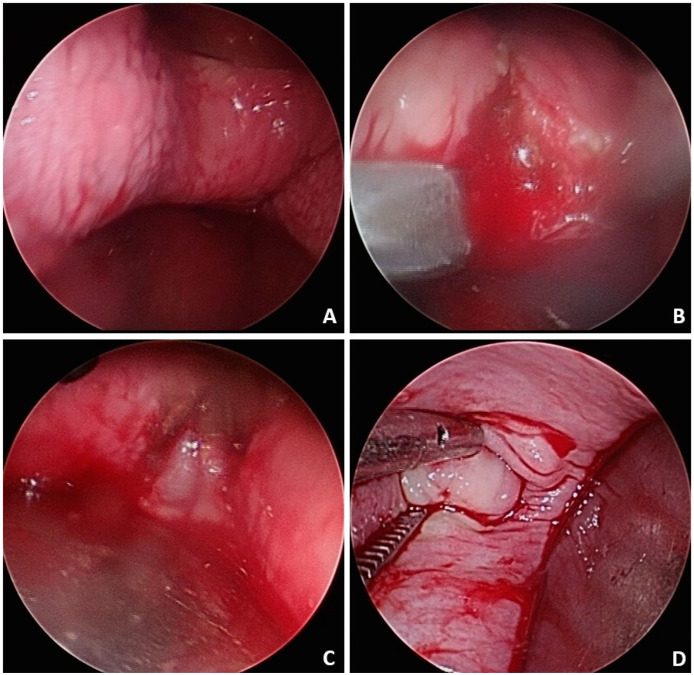
(**A**) The engorged right vaginal wall; (**B**) a speculum was inserted into the vagina for excision of the diverticulum; (**C**) pus with hematoma in the right vaginal diverticulum; (**D**) the marsupialization of the right vaginal wall.

## Data Availability

Data sharing is not applicable to this article.

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
