# Peer review of "Herlyn–Werner–Wunderlich Syndrome Complicated with Vesicovaginal Fistula: A Rare Case Report"

_medicina, 2024, doi:10.3390/medicina60071081_

Round 1
Reviewer 1 Report
Comments and Suggestions for Authors
Dear Authors,
Obstructed hemivagina and ipsilateral renal agenesis (OHVIRA) syndrome, previously known as "Herlyn-Werner-Wunderlich Syndrome," is a rare congenital defect of the Mullerian ducts.
The manuscript is interesting but lacks novelty. At least 10 cases have been published in the last five years. To improve the manuscript, I suggest the authors add a review to their presentation.
Abstract - please include the final treatment.
l. 32-37 please include references
l. 58 did the authors performed pelvis ultrasound?
Figures - please include more images, and add some arrows for a better understanding
What happens to the vesicovaginal fistula?
Please include imaging aspects after the final treatment.
The Discussion section needs to be revised, regarding the treatment.
Comments on the Quality of English LanguageThe English is fine
Reviewer 2 Report
Comments and Suggestions for Authors
Dear Authors
-Your case is interesting and rare but there are many similar cases reported in the literature, what was the main reason you reported it?
-please explain more technical details of your procedure.
-Did you perform vaginoscopy?
-It would be more attractive if you added pictures of what you faced during the operation and a short video of the procedure.
Round 2
Reviewer 2 Report
Comments and Suggestions for Authors
Dear Authors
Thanks for your response. It is better to add short terms followup of your case.
Author Response
Comment : [Thanks for your response. It is better to add short terms followup of your case.]
Author Reply:
Thank you for your valuable and practical comment. The patient had a good postoperative recovery, and was discharged successfully on the second day after surgery. After a 3-month follow-up, the patient had a complete recovery without any sequelae. We had revised our abstract and case presentation in manuscript, and uploaded it to Medicina.
